# COVID-19 mRNA vaccine induced antibody responses against three SARS-CoV-2 variants

Pinja Jalkanen [1✉], Pekka Kolehmainen [1], Hanni K. Häkkinen[2], Moona Huttunen [1], Paula A. Tähtinen[3], Rickard Lundberg[1], Sari Maljanen[1], Arttu Reinholm[1], Sisko Tauriainen [1], Sari H. Pakkanen[2], Iris Levonen[2], Arttu Nousiainen[2], Taru Miller[2], Hanna Välimaa[2,4], Lauri Ivaska [3], Arja Pasternack[5], Rauno Naves[5], Olli Ritvos[5], Pamela Österlund [6], Suvi Kuivanen[4], Teemu Smura[4], Jussi Hepojoki [4], Olli Vapalahti[4], Johanna Lempainen[1,3], Laura Kakkola[1,8], Anu Kantele[2,8] & Ilkka Julkunen [1,7,8✉]

As SARS-CoV-2 has been circulating for over a year, dozens of vaccine candidates are under development or in clinical use. The BNT162b2 mRNA COVID-19 vaccine induces spike protein-specific neutralizing antibodies associated with protective immunity. The emergence of the B.1.1.7 and B.1.351 variants has raised concerns of reduced vaccine efficacy and increased re-infection rates. Here we show, that after the second dose, the sera of BNT162b2-vaccinated health care workers (n = 180) effectively neutralize the SARS-CoV-2 variant with the D614G substitution and the B.1.1.7 variant, whereas the neutralization of the B.1.351 variant is five-fold reduced. Despite the reduction, 92% of the seronegative vaccinees have a neutralization titre of >20 for the B.1.351 variant indicating some protection. The vaccinees' neutralization titres exceeded those of recovered non-hospitalized COVID-19 patients. Our work provides evidence that the second dose of the BNT162b2 vaccine induces cross-neutralization of at least some of the circulating SARS-CoV-2 variants.

[1] Institute of Biomedicine, University of Turku, Turku, Finland. [2] Department of Infectious Diseases, Meilahti Vaccination Research Center, MeVac, Helsinki University Hospital and University of Helsinki, Helsinki, Finland. [3] Department of Paediatrics and Adolescent Medicine, Turku University Hospital and University of Turku, Turku, Finland. [4] Department of Virology, University of Helsinki, Helsinki, Finland. [5] Department of Physiology, University of Helsinki, Helsinki, Finland. [6] Finnish Institute for Health and Welfare, Helsinki, Finland. [7] Clinical Microbiology, Turku University Hospital, Turku, Finland. [8] These authors jointly supervised: Laura Kakkola, Anu Kantele, Ilkka Julkunen. ✉email: pinja.r.jalkanen@utu.fi; ilkka.julkunen@utu.fi

The emergence and spread of the severe acute respiratory syndrome coronavirus 2 (SARS-CoV-2) has caused a pandemic with over 3.8 million deaths[1] and rapid development of multiple vaccine candidates[2]. SARS-CoV-2 infection elicits antibodies against spike protein (S) and nucleoprotein (N)[3–5], of which, on the basis of virus challenge studies in animals, the spike protein-specific antibodies are neutralizing and associated with protective immunity[6,7]. In addition, recent studies of COVID-19 patients and vaccinees indicate that previous infections and vaccinations are related to a decreased rate of SARS-CoV-2 infections[8–10]. Although the persistence of vaccine-induced antibodies is still not known, infection-induced neutralizing antibodies have remained detectable for at least six months after symptom onset[11].

Currently, European Medicines Agency (EMA) has authorized four vaccines to be used in European Union: two mRNA vaccines (BNT162b2/Comirnaty by Pfizer-BioNTech and mRNA-1273 by Moderna) and two adenoviral vector-based vaccines (ChAdOx1-S by AstraZeneca-Oxford and COVID-19 Vaccine Janssen by Janssen Biologics B.V. and Janssen Pharmaceutica NV)[12]. All four vaccines aim to generate spike protein-specific antibodies and all have been shown to induce anti-S IgG antibodies with neutralizing activity against the first pandemic SARS-CoV-2 Wuhan Hu-1 variant and the currently circulating D614G variants[13–15]. The recent emergence of SARS-CoV-2 variants of concern, such as B.1.1.7 first identified in the United Kingdom[16] and B.1.351 first identified in South Africa[17], has raised concerns about increased virus transmissibility and reduced vaccine efficacy. These two variants of concern are defined by eight to ten amino acid changes or deletions in the spike protein to which vaccine-induced antibodies are targeted[17–20]. Both of these variants are now transmitted in several countries (https://cov-lineages.org/global_report.html). Initial studies reported that antibodies produced in response to vaccination and natural infection neutralize the B.1.1.7 variant[19,21], whereas neutralization of the B.1.351 is reduced 8–13-fold[18,22,23]. However, it is still unclear whether the B.1.351 variant can escape from humoral and cell-mediated immunity.

Here, we characterize the BNT162b2 vaccine-induced antibody responses among a sequential serum sample cohort of 180 Finnish healthcare workers who, belonging to the group vaccinated first in Finland, received two doses of COVID-19 vaccine with three weeks interval. SARS-CoV-2 S1-specific IgG, IgA, and IgM antibody responses and neutralization titres for three SARS-CoV-2 variants were determined. We show that two-dose immunization yields high levels of anti-S1 IgG antibodies in 100% of vaccinees. The second vaccine dose induces antibodies for efficient neutralization of D614G and B.1.1.7. variants, whereas the neutralization titres for B.1.351 are lower.

vaccination on 28 December 2020, and the last vaccinations for those HCWs in this study were given on 12 February 2021. Serum samples from 50 non-hospitalized, recovered COVID-19 patients from 2020 were also included in the analysis. The generation of anti-S1 IgG, IgA, IgM, and total Ig antibodies after vaccination was analyzed with enzyme immunoassay (EIA). The original optical density values in the assay were converted to EIA units to minimize inter-assay variation (Fig. 1A). To verify the EIA results and to study the rate of SARS-CoV-2 infections after vaccination, sera were also analyzed with N protein-specific EIA (Fig. 1B).

Before vaccination (0 day sampling) 11/180 (6%) had anti-S1 IgG antibodies (Supplementary Fig. 1A) indicating that these individuals had undergone a previous SARS-CoV-2 infection. Five of these anti-S1 positive participants had also anti-N IgG antibodies. Already after the first dose of the vaccine (3 weeks), vaccinees with prior SARS-CoV-2 infection showed clearly increased levels of anti-S1 IgG antibodies (geometric mean 99). After the second dose of the vaccine (6 weeks), all vaccinees with prior SARS-CoV-2 infection had very high levels of anti-S1 IgG antibodies (geometric mean 109) (Supplementary Fig. 1A).

Three weeks after the first dose of BNT162b2, vaccinees without prior SARS-CoV-2 infection (169/180, 94%) developed varying levels of anti-S1 IgG antibodies (geometric mean 47), and moderate levels of anti-S1 IgA and IgM antibodies (Fig. 1A). Total Ig levels for S1 ranged from 1 to 98 EIA units. The EIA levels for anti-N antibodies among the vaccinees remained the same as before the vaccination (Fig. 1B, Table 1) indicating the absence of SARS-CoV-2 infections in this study group. Anti-N IgG antibody levels were higher in sera of non-hospitalized COVID-19 patients (geometric mean 17) than of vaccinees (geometric mean 2) (Fig. 1B, Table 1). However, already after the first vaccination dose, the geometric mean of anti-S1 IgG and anti-S1 total Ig antibodies of vaccinees exceeded those of convalescent-phase COVID-19 patients, 47 and 37 vs. 20 and 23, respectively (Fig. 1A, Table 1).

Six weeks after the first vaccine dose (3 weeks after the second vaccine dose) all participants, despite the initial response after the first vaccine dose, elicited high levels of anti-S1 IgG antibodies together with a modest increase in anti-S1 IgA and IgM antibody levels (Fig. 1A). Based on anti-N antibodies, only one person was infected with SARS-CoV-2 during the 6 weeks: the participant was anti-N and anti-S1 IgG negative at 0-day sampling, and anti-N and anti-S1 IgG positive at 3-week sampling. The negative control antigen signals were close to the background values (Supplementary Fig. 2). Anti-S1 IgG antibody and total anti-S1 Ig levels induced by two doses of BNT162b2 vaccine were clearly higher than the anti-S1 IgG levels measured from the convalescent-phase patient sera, geometric means being 107 and 86 vs. 20 and 23, respectively.

## Results

**Study subjects**. The vaccinee group comprised 180 volunteers (115 from Turku University Hospital, TUH and 65 from Helsinki University Hospital, HUH), aged 20–65 years (mean age 43 and median 41); 149/180 (83%) were females (age 20–65 years) and 31/180 (17%) were males (age 22–60 years). The group of recovered COVID-19 patients comprised 50 volunteers (from HUH), aged 19–93 (mean 43 and median 38); 33 were females and 17 males.

**Antibody responses against SARS-CoV-2 S1 and N proteins in vaccinees and convalescent-phase patients**. In order to monitor the immunological responses of vaccinees, we collected sequential serum samples (0, 3, 6 weeks) from 180 vaccinated health care workers (HCWs). The first vaccinated HCWs received their first

**Characterization of SARS-CoV-2 isolates**. To analyze the neutralization capacity of the vaccinees' sera, we isolated for microneutralization tests four virus variants circulating in Finland: D614G variants FIN-25 (spring 2020) representing B.1 lineage and SR121 (autumn 2020) representing B.1.463 lineage, a variant of concern 85HEL representing B.1.1.7 lineage and a variant of concern HEL12-102 representing B.1.351 lineage. FIN-25 isolate was passaged first in VeroE6 cells followed by passaging in VeroE6 cells expressing transmembrane protease serine 2 (VeroE6-TMPRSS2-H10). Other three isolates were passaged in VeroE6-TMPRSS2-H10 cells to avoid the generation of mutations in the vicinity of the furin cleavage site (Fig. 2A). The isolates were sequenced to compare the mutations in SR121, 85HEL (B.1.1.7), and HEL12-102 (B.1.351) variants to FIN-25 that represented the circulating strains in Finland until the emergence of variants of concern. Sequence analysis of SARS-CoV-2

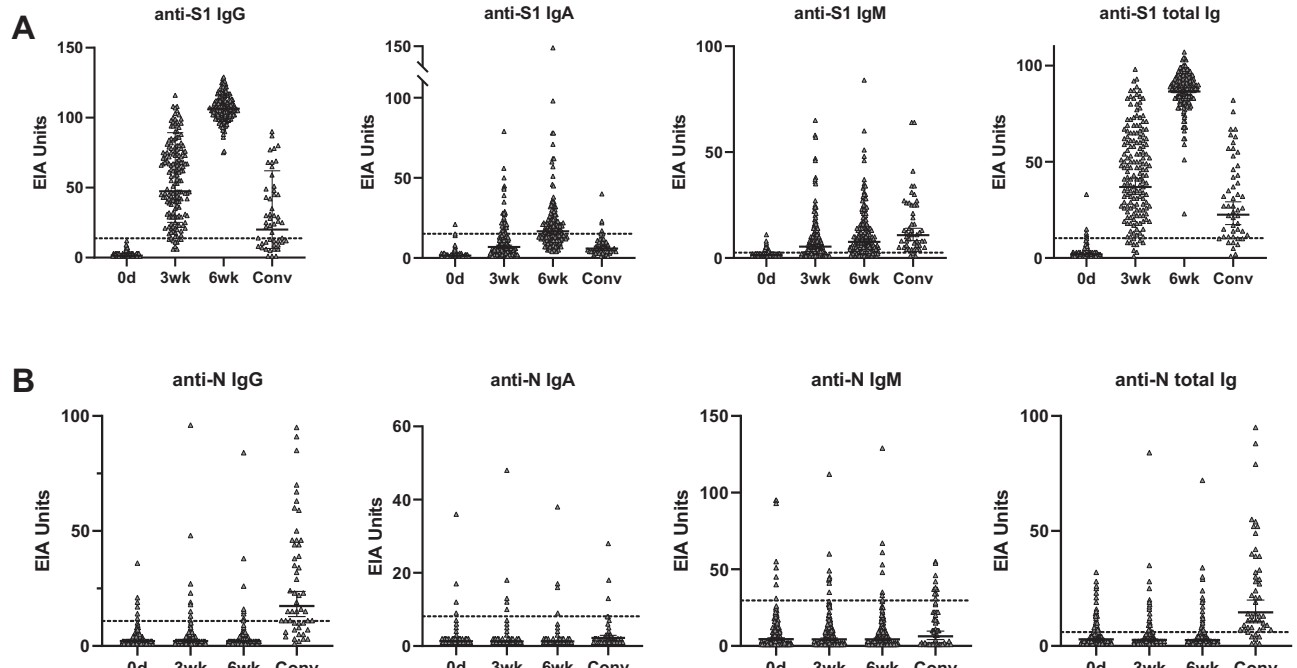

**Fig. 1 Antibody responses against SARS-CoV-2 S1 and N proteins in BNT162b2 vaccinated health care workers and non-hospitalized recovered COVID-19 patients. A** Anti-S1 and **B** anti-N IgG, IgA, IgM, and total Ig antibody levels were measured with EIA. Serum samples from BNT162b2 vaccinated initially seronegative participants (n = 169) were collected before vaccination (0d), and three (3wk) and six (6wk) weeks after the first dose of the vaccine. All vaccinees received the second dose of the vaccine three weeks after the first dose. Convalescent phase patient samples (Conv, n = 50) were collected 14 days–6 weeks after the positive RT-qPCR test result. Data are represented as geometric means and geometric standard deviations (SD). Cut-off values are indicated with dashed lines.

**Table 1 Antibody responses in BNT162b2 vaccinated health care workers (HCW) without previous SARS-CoV-2 infection and non-hospitalized convalescent-phase COVID-19 patients.**

| | | 0d | | 3wk | | 6wk | | Convalescent | |
|---|---|---|---|---|---|---|---|---|---|
| | | GM (95% CI) | Positive (n/n) | GM (95% CI) | Positive (n/n) | GM (95% CI) | Positive (n/n) | GM (95% CI) | Positive (n/n) |
| EIA | Anti-S1 IgG | 1 (1.4–1.7) | 0% (0/169) | 47 (43–52) | 96% (160/167) | 107 (105–108) | 100% (169/169) | 20 (15–28) | 62% (31/50) |
| | Anti-S1 tot Ig | 2 (1.7–2.2) | 4% (6/169) | 37 (33–41) | 96% (160/167) | 86 (85–88) | 100% (169/169) | 23 (17–29) | 82% (41/50) |
| | Anti-N IgG | 2 (1.9–2.4) | 4% (6/169) | 2 (1.9–2.5) | 7% (11/167) | 2 (1.8–2.4) | 5% (9/169) | 17 (13–23) | 66% (33/50) |
| MNT | FIN-25 | 10 (10–10) | 0% (0/169) | 24 (21–28) | 63% (106/167) | 234 (210–261) | 100% (169/169) | 55 (42–73) | 86% (43/50) |
| | SR121 | 10 (10–10) | 0% (0/84) | 32 (27–37) | 83% (70/84) | 275 (234–323) | 100% (86/86) | 86 (67–110) | 96% (48/50) |
| | 85HEL | 10 (10–10) | 0% (0/169) | 24 (21–28) | 63% (106/167) | 240 (214–269) | 100% (169/169) | 74 (58–93) | 96% (48/50) |
| | HEL12-102 | 10 (10–10) | 0% (0/169) | 12 (11–13) | 15% (25/167) | 48 (45–54) | 92% (156/169) | 16 (14–18) | 56% (28/50) |

HCW serum samples (n = 169) were collected before vaccination (0d), and three (3wk) and six (6wk; three weeks after the second vaccine dose) weeks after the first vaccine dose. Geometric means (GM), 95% confidence intervals (CI) and the number of positive samples for anti-S1 IgG and total Ig, and anti-N IgG antibodies and neutralizing antibodies is indicated. In microneutralization test (MNT) neutralization titre 20 or higher was considered positive and for calculation of geometric means a value of 10 was given for values of <20.

isolates revealed 3 amino acid changes in the spike protein of FIN-25, 4 in SR121, 10 in 85HEL (B.1.1.7), and 9 in HEL12-102 (B.1.351) variants compared to the original Wuhan Hu1 strain (Fig. 2A). The sequence of FIN-25 that was passaged initially in VeroE6 cells had close to the furin cleavage site a deletion of amino acids 674–678 in 45% and R682W mutation at the furin cleavage site in 41% of the virus population, indicating some heterogeneity of the FIN-25 virus stock, which did, however, not affect the growth properties of the

virus. Sequences of the three other isolates passaged in VeroE6-TMPRSS2-H10 cells only had either aforementioned deletion in the minority of the virus population (8% of SR121) or a completely intact furin cleavage site. Otherwise, all spike protein sequences obtained from the virus propagations were identical to the sequences obtained from the respective original patient sample, all also contained the D614G substitution linked to increased fitness and transmissibility[24,25].

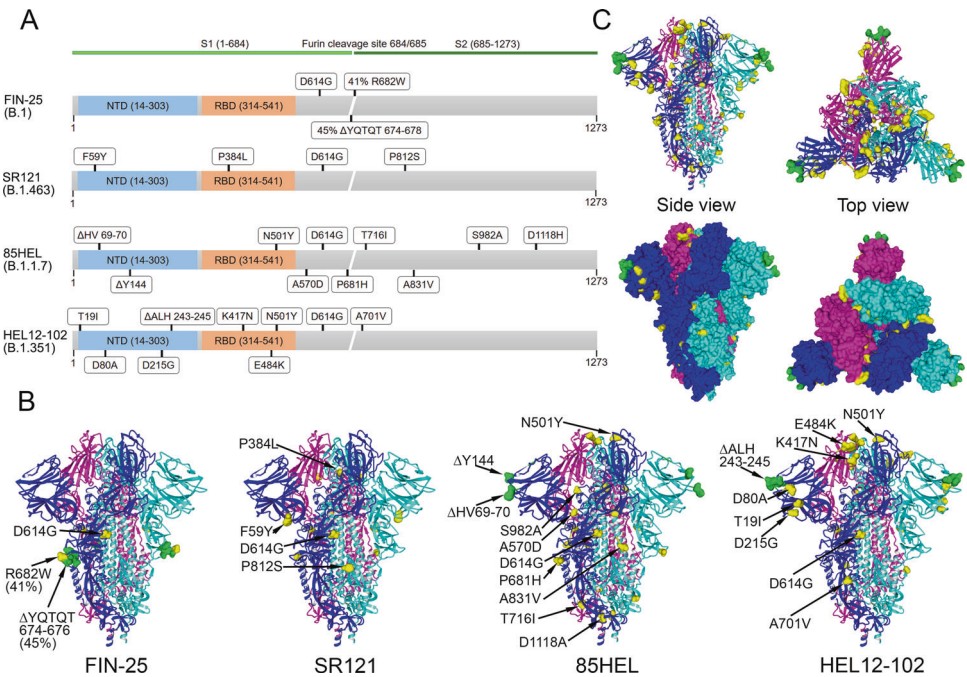

**Fig. 2 Genetic analysis of virus variants and spike protein structure prediction. A** Schematic presentation of S gene and amino acid changes in FIN-25 (B.1 lineage), SR121 (B.1.463), 85HEL (B.1.1.7), and HEL12-102 (B.1.351) virus isolates used in the present study. **B** Trimeric SARS-CoV-2 S protein, referred to as the spike, structure in the closed conformation (pdb: 6VXX). Amino acid substitutions (yellow) and deletions (green) as compared to the original spike structure. **C** Collective presentation of all amino acid changes found in virus isolates. Space-filling model indicating amino acids changes (yellow) and deletions (green) on the surface of a trimeric S protein. Side and top views are shown.

The availability of the 3-dimensional structure of the SARS-CoV-2 spike protein enabled the positioning of the amino acid changes into the structure of the trimeric spike protein (Fig. 2B). Substitutions found in FIN-25 and SR121 spike proteins localize in the stem regions of the trimeric spike protein. The substitutions found in the spike proteins of B.1.1.7 and B.1.351 variants located both to the stem region and on the surface of the trimeric spike protein close to the receptor-binding domain (RBD). The three B.1.351 variant substitutions E484K, K417N, and N501Y are in the groove of the RBD–ACE2 interaction domain. In addition, both the B.1.1.7 and B.1.351 variants had three amino acid deletions in the far edges of the 3-dimensional structure (Fig. 2B). Figure 2C shows combined amino acid changes found in the isolates used in this study indicating the accumulation of substitutions on multiple localizations on the trimeric structure of the spike protein. The amino acid changes in the spike protein, especially the aforementioned E484K, K417N, and N501Y have recently been reported to affect the neutralizing efficacy of the antibodies[26].

**Neutralizing antibody titres against SARS-CoV-2 variants.** To measure the neutralizing potential of the vaccinees' sera against all four SARS-CoV-2 isolates, neutralizing antibody titres elicited by the BNT162b2 vaccine were analyzed with microneutralization test (MNT). The neutralizing titres with two D614G isolates FIN-25 and SR121 were almost identical both 3 weeks ($p = 0.02$) and 6 weeks after the first dose ($p = 0.11$) (Fig. 3A), indicating that the mutations in FIN-25 spike protein due to initial propagation in VeroE6 cells did not affect the neutralizing titres.

Before vaccination (0 day sampling) those 11/180 with a likely previous SARS-CoV-2 infection based on EIA results, showed increased geometric mean titres (GMT) of 35, 31, and 16 against FIN-25, 85HEL (B.1.1.7), and HEL12-102 (B.1.351) virus isolate, respectively. Following the first and the second vaccination, the GMTs further increased to 435 and 682, to 320 and 640, and to 101 and 132, respectively (Supplementary Fig. 1B). These results with this small group indicate that even one dose of the BNT162b2 vaccine induces high MNT titres in those individuals who had suffered a previous COVID-19 infection.

None of the vaccinees without a prior SARS-CoV-2 infection (169/180) had neutralizing antibodies before the vaccination (Fig. 3A). Three weeks after the first vaccine dose, neutralizing titres against all four isolates were slightly increased (GMT of 24 for FIN-25, 32 for SR121, 24 for 85HEL (B.1.1.7), and 12 for HEL12-102 (B.1.351)). Six weeks after the first dose of the vaccine (3 weeks after the second dose), neutralizing titres were increased to a GMT of 234 against FIN-25, 275 against SR121, 240 against 85HEL (B.1.1.7), and 48 against HEL12-102 (B.1.351) (Fig. 3A, Table 1). Three weeks after the first dose 37%, 17%, 37%, and 85% of vaccinees had a neutralization titre <20 against FIN-25, SR121, 85HEL (B.1.1.7), and HEL-12-102 (B.1.351) isolates, respectively. After the second vaccine dose, 100% of vaccinees had neutralizing antibodies against FIN-25, SR121 and 85HEL (B.1.1.7), whereas 92% of vaccinees had neutralizing antibodies against the HEL-12-102 (B.1.351) variant. GMTs against all four isolates in vaccinees exceeded the GMTs seen in convalescent-phase patient sera (Fig. 3A, Table 1).

Three weeks after the first vaccine dose, the GMT for HEL-12-102 (B.1.351) was 2-fold lower ($p < 0.0001$) compared to FIN-25 and 85HEL (B.1.1.7). After the second immunization, the GMT for HEL-12-102 (B.1.351) was 5-fold lower compared to FIN-25 and 85HEL (B.1.1.7) (Fig. 3B).

The MNT titres for two D614G-containing isolates, FIN-25 and SR121, correlated very well, as also did FIN-25 and 85HEL (B.1.1.7) ($r > 0.8$, $p < 0.0001$) (Fig. 4). MNT titres for FIN-25 and HEL-12-102 (B.1.351) correlated relatively well and highly significantly ($r = 0.74$, $p < 0.0001$), as did the two variants of concern, 85HEL (B.1.1.7) and HEL-12-102 (B.1.351) ($r = 0.75$, $p < 0.0001$).

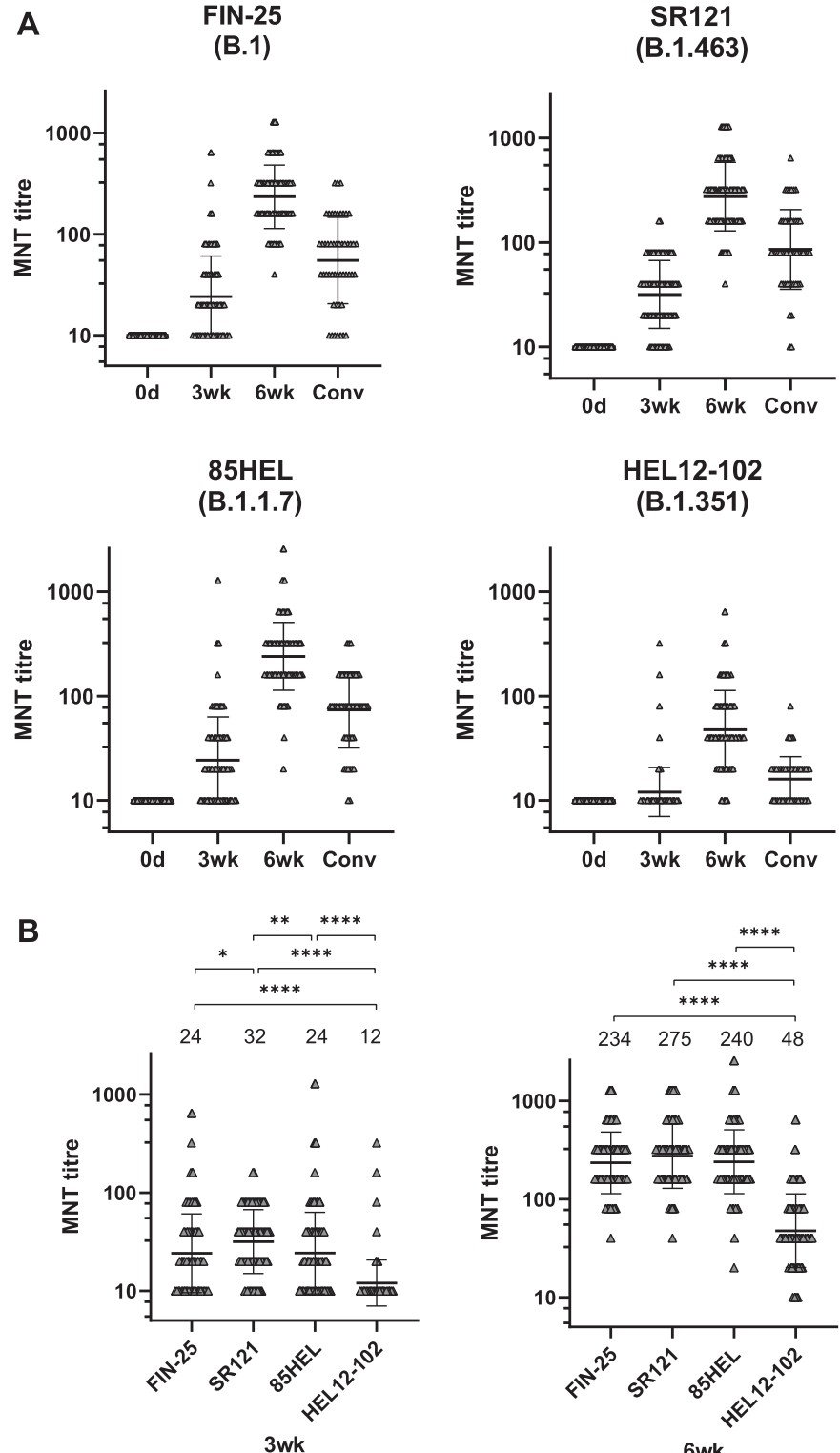

**Fig. 3 Neutralization of B.1.1.7 and B.1.351 variants by BNT162b2 vaccinees' sera and COVID-19 patient sera. A** Neutralization titres of initially seronegative vaccinees (*n* = 169) for D614G variants FIN-25 and SR121, and 85HEL (B.1.1.7) and HEL12-102 (B.1.351) variants before (0d), three (3wk), and six weeks (6wk) after the first dose of BNT162b2 vaccine and neutralization titres of convalescent sera of non-hospitalized patients (Conv, *n* = 50). Values above the groups indicate geometric mean titres (GMTs) and data are shown as geometric means and geometric SDs. Neutralization titres <20 were plotted as 10. **B** Neutralization titres 3 weeks (3wk) and six weeks (6wk) after the first dose of the vaccine. Statistical differences between the virus isolates were analyzed with Wilcoxon matched-pairs signed-rank test. Two-tailed *p*-values <0.05 were considered significant. Exact *p*-values were *=0.0201, **=0.0015, ****<0.0001 for 3wk and 6wk. Values above the groups indicate geometric mean titres (GMTs) and data are shown as geometric means and geometric SDs. Neutralization titres <20 were plotted as 10.

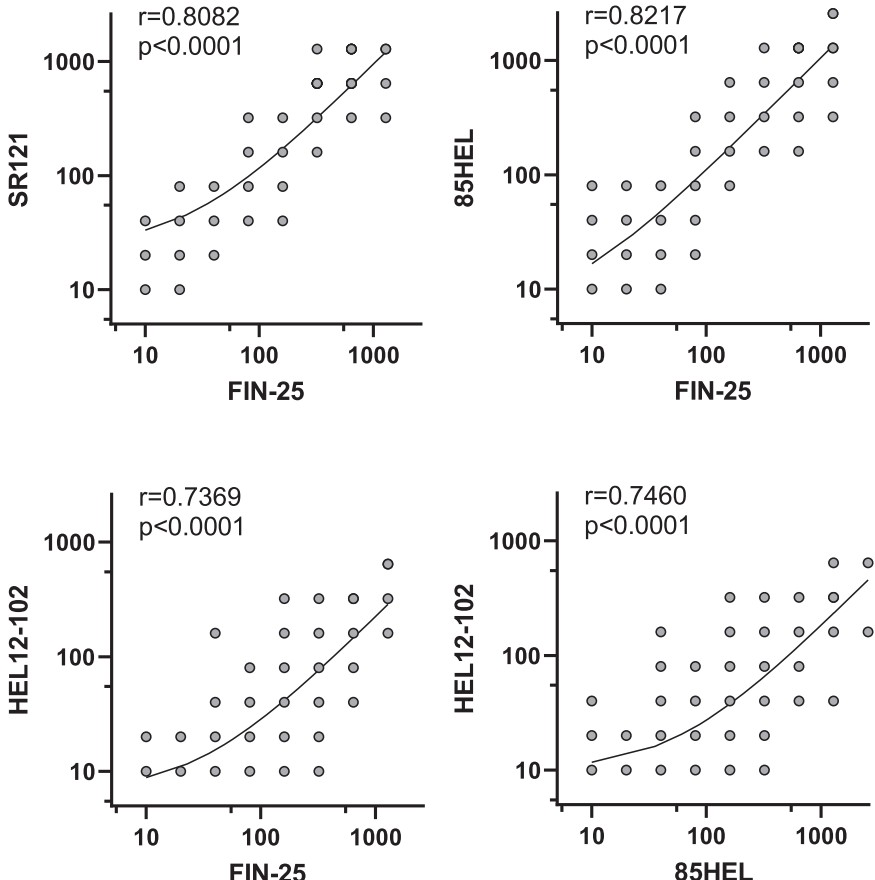

**Fig. 4 Correlation of MNT titres against SARS-CoV-2 isolates.** MNT titres of BNT162b2 vaccinees (initially seronegative, $n = 169$) against FIN-25 were compared with MNT titres against 85HEL (B.1.1.7) and HEL12-102 (B.1.351) variants including 0d, 3wk, and 6wk samples. Comparison between two D614G virus isolates, FIN-25 and SR121, was done with sera from 86 BNT162b2 vaccinees. Correlation co-efficient ($r$) was calculated with Pearsons correlation test and two-tailed $p$-values < 0.05 were considered significant. Each dot may represent multiple samples.

To analyze the effect of age and gender on the antibody responses, the vaccinees were divided into age and gender groups, and the S1 IgG EIA and MNT results were compared between the groups (Fig. 5A and B). After the first vaccine dose, anti-S1 IgG antibody levels and neutralization titres were significantly lower in the older age group (55–65 years) compared to younger age groups (20–34 and 35–44 years) (Fig. 5A). However, after the second vaccine dose, the neutralization titres were similar between the age groups (GMT 257, 268, 200, and 206 in age groups of 20–34, 35–44, 45–54, and 55–65 years, respectively) (Fig. 5A). We also compared gender-related antibody responses even though male vaccinees were underrepresented, comprising only 17% (29/169) of the vaccinees. After the second dose, female vaccinees had slightly higher neutralization titres than males ($p = 0.0412$), although the anti-S1 IgG antibody levels remained at the same level (Fig. 5B).

**EIA values correlate with MNT titres.** Neutralization tests with live SARS-CoV-2 viruses are very time-consuming, and at the moment the assay requires BSL-3 laboratory conditions, whereas EIA and other similar colorimetric/fluorometric antibody assays are faster and user-friendlier. To assess whether EIA values are associated with MNT titres, anti-S1 IgG and total anti-S1 Ig were compared to neutralization titres against FIN-25 (Fig. 6, Supplementary Fig. 3). Both anti-S1 IgG and total anti-S1 Ig EIA measurements correlated very well with MNT titres ($r > 0.9$, $p < 0.0001$) suggesting that EIA, especially IgG EIA, using spike protein as an antigen can be a useful method to determine COVID-19 immunity.

**Discussion**

The emergence of the COVID-19 pandemic in early 2020 prompted a rapid development of various types of vaccines such as mRNA encoding SARS-CoV-2 spike protein, viral vector-based (e.g. adenovirus), inactivated virus, virus-like particle, and recombinant protein vaccines. Once the European Union had made agreements with a number of vaccine producers, mass immunization was started in Finland at the end of December 2020, first with the mRNA-based Pfizer-BioNTech vaccine and somewhat later the Moderna mRNA and AstraZeneca adenovirus-based vaccines[12]. Vaccination of health care professionals within a national vaccination programme in Finland enabled us to start, independent of pharmaceutical companies, a follow-up study of vaccine-induced immunity. In the present report, we show that two-dose vaccination with the BNT162b2 mRNA COVID-19 vaccine induces very high antibody levels against viral spike protein and high titres of neutralizing antibodies. The vaccine induced good cross-reactivity to D614G and B.1.1.7 variants in all vaccinees and, albeit reduced levels, detectable neutralizing antibodies to B.1.351 variant in 92% of the vaccinees.

EIA is a rapid and sensitive method to analyze immune responses against vaccine antigens or different viral proteins in response to infection. The method is easily quantitative and suitable for analyzing different immunoglobulin classes. In this study, we observed that practically all seronegative health care workers (20–65 years of age) responded to the first BNT162b2 vaccine dose and an increase in spike protein-specific antibody

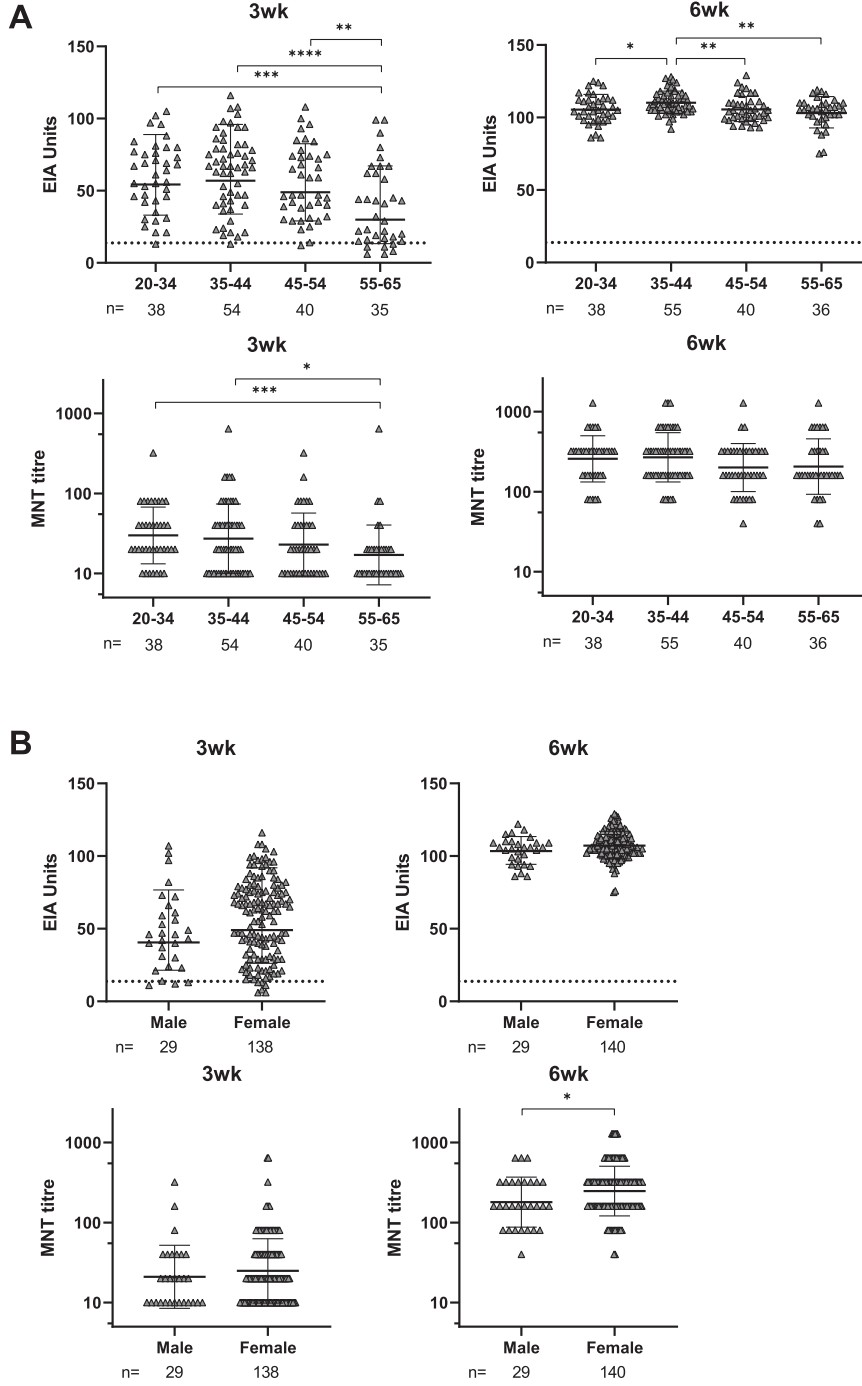

**Fig. 5 Antibody responses against SARS-CoV-2 S1 protein and neutralization of FIN-25 by age and gender. A** BNT162b2 vaccinated health care workers (initially seronegative, $n = 169$) were divided into four age groups. Age specific differences of anti-S1 IgG antibody levels and neutralization titres against FIN-25 virus isolate were analyzed. Sera was collected three weeks (3w) and six weeks (6wk) after the first vaccine dose. Differences between age groups were tested with two-tailed Mann–Whitney $U$ test. Two-tailed $p$-values < 0.05 were considered significant. Exact $p$-values were **=0.0078, ***=0.0007, and ****<0.0001 for 3wk EIA, **=0.0021 (age group 35–44 vs. 55–65), *= 0.0231, and **=0.0041 for 6wk EIA, and *=0.0133 and ***=0.0005 for 3wk MNT. **B** Gender-specific differences in S1 specific IgG antibody responses and neutralization titres against FIN-25 were analyzed. Differences between age and gender groups were tested with two-tailed Mann–Whitney $U$ test. Two-tailed $p$-values <0.05were considered significant. Exact $p$-values were *=0.0412. The data in **A** and **B** are presented as geometric means and geometric SDs. Neutralization titres <20 were plotted as 10.

responses in the IgG antibody class was detectable. The IgG antibody levels varied considerably and relatively few individuals showed increased antibody levels in the IgA and IgM antibody classes. The second vaccine dose, which was given according to the original vaccination protocol 3 weeks after the first vaccine dose, induced very high levels of spike protein-specific IgG

antibodies, while IgA and IgM responses remained low. The vaccinees' IgG antibody levels were on average higher than those seen in convalescent-phase sera of home-treated patients. Anti-body responses have been found generally higher for COVID-19 patients with a more severe disease[23,27,28], however, as shown by this study also, the BNT162b2 vaccine appears to induce higher

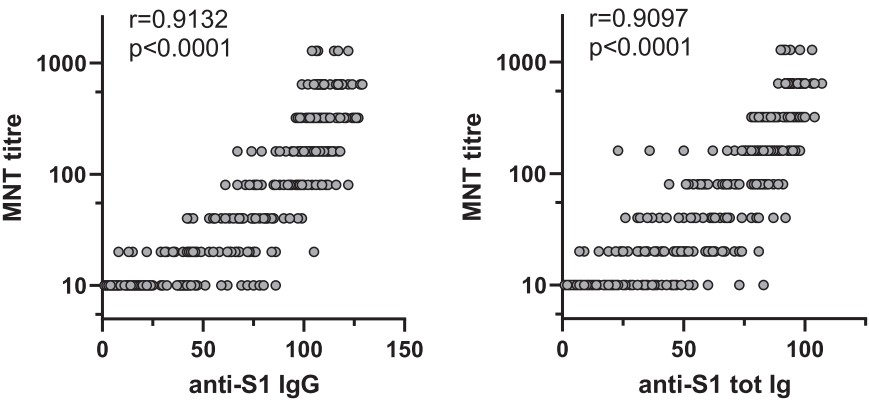

**Fig. 6 Correlation of anti-S1 antibody levels with SARS-CoV-2 neutralization titres.** Anti-S1 IgG and total Ig antibody levels were determined with EIA and neutralization titres of BNT162b2 vaccinated health care workers (initially seronegative, $n = 169$) against FIN-25 virus isolate were obtained with microneutralization test (MNT). All sequential serum samples (0d, 3wk, and 6wk) were included in the analysis. Spearman's rank correlation coefficient ($r$) is indicated.

antibody levels than those measured in patients[29,30]. Remarkably, the administration of two doses of the mRNA vaccine induced very high antibody responses in 100% of the vaccinees.

The global circulation of SARS-CoV-2 and a huge number of infections worldwide have led to the emergence of hundreds of evolutionary lineages and variants of the virus (https://cov-lineages.org/global_report.html). The evolutionary speed of SARS-CoV-2 has been relatively slow, at least compared to influenza A viruses, presumably due to a virus-encoded enzyme with a proof-reading capability. Within the first 16 months of circulation, up to 30–35 mutations have been identified accumulating into the viral genome. Many of these mutations are silent or appear in places of the genome that are not critical for avoiding immunity induced by vaccination or natural infection. However, a number of variants have raised concern due to mutations accumulating particularly in the S-gene and causing changes in the immunodominant epitopes of the trimeric spike protein. Mapping the spike protein mutations on variants sequenced and used in this study revealed that they occur outside the globular head of the trimeric spike protein. The D614G and B.1.1.7 variant viruses were readily neutralized by the vaccinees' sera, indicating that these mutations are unlikely to impair the neutralizing antibody capacity induced by vaccination or natural infection. However, it should be noted that the neutralizing titre of these sera was five-fold lower against the B.1.351 variant, which denotes that the amino acid changes accumulating in this variant are potentiating the escape of the virus from the humoral immune responses. Despite this, more than 92% of the vaccinees showed measurable neutralizing antibody titres against the B.1.351 variant, suggesting that the spike protein encoded by Pfizer-BioNTech's mRNA vaccine is similar enough to also mount an immune response against the B.1.351 variant.

The critical amino acid changes linked to escape from humoral immunity in the B.1.351 variant appear to be K417N, E484K, and N501Y[30–32]. These amino acids are situated in the grooves within the receptor-binding site of the trimeric S protein complex. There is no three-dimensional structure presently available for the B.1.351 variant spike protein trimer, but because of its relatively radical amino acid substitutions, conformational changes in the spike structure may prove substantial. Interestingly, the B.1.351 and B.1.1.7 variants have deletions in the tips of the globular S1 domain (amino acids 243–245 and amino acids 69–70 and 244, respectively) which could contribute to the impaired recognition by neutralizing antibodies.

It is currently not known how high neutralizing antibody titres against a given virus variant are required for antibody-mediated protection against the COVID-19. However, the clinical efficacy

data accumulating from COVID-19 vaccine studies strongly suggest that already one dose of the vaccine provides protection against severe COVID-19, even when neutralizing antibody levels cannot be detected in all vaccinees[33,34]. This suggests that the first vaccine dose may prime the individual for rapid induction of protective immunity when contracting the virus in nature and avoiding severe COVID-19. According to previous data[29,35–38], we found that individuals with prior SARS-CoV-2 infection readily responded to the first vaccine dose with high antibody levels and neutralization titres.

Humoral immune response to vaccinations has been shown to decline with age[39,40]. Consistently, we observed a trend of declining immune response to the COVID-19 mRNA vaccine by age. This trend was not very strong, presumably because the ages of our vaccinees ranged from 20 to 65 years, while age-dependent immunosenescence should be more pronounced in the age group >65 years[39]. Another explanation might be that the BNT162b2 mRNA vaccine is exceptionally immunogenic and therefore, especially when given two doses, it enables practically all individuals regardless of gender and age, to develop high antibody levels and neutralization titres.

In summary, in the present study we show that the Pfizer-BioNTech BNT162b2 COVID-19 mRNA vaccine is highly immunogenic, and particularly after two vaccine doses, all vaccinees showed a very high humoral immune response to D614G variant viruses. Immunity to a recent B.1.1.7 variant was equally good as compared to the D614G variant, whereas vaccine and SARS-CoV-2 infection induced immunity against B.1.351 variant was reduced. Despite this, almost all vaccinees showed neutralizing antibodies against the B.1.351 variant, suggesting to provide at least some degree of protection against these variant viruses. In the future, it will be intriguing to study the development and persistence of cell-mediated immunity induced by COVID-19 vaccines. Promising data have been reported at least for the BNT162b2 vaccine which in preliminary studies has induced good cell-mediated immunity[41,42]. As the use of other types of SARS-CoV-2 vaccines will be increased, it is the responsibility of the scientific community and public health professionals to systematically collect serum and cellular samples for comparative analyses of vaccine-induced immunity, cross-protection, and longevity of vaccine and natural infection-induced immunity.

As a whole, all vaccines that have currently obtained market authorization in the EU show excellent protective efficacy against severe COVID-19. Thus, it is very likely that immunogenicity results similar to those presented here will be applicable to them as well.

## Methods

**Study participants**. SARS-CoV-2 vaccinations started in Finland at the end of December 2020 with Pfizer-BioNTech BNT162b2 mRNA (Comirnaty) vaccine. Study participants ($n = 180$) were recruited among healthcare personnel of Turku University Hospital (TUH, Turku, Finland) (Southwest Finland health district ethical permission ETMK 19/1801/2020) and Helsinki University Hospital (HUH, Helsinki, Finland) (Helsinki-Uusimaa health district ethical permission HUS/1238/2020) prior to receiving an optimal regimen of two doses of BNT162b2 mRNA vaccine at a 3-week dosing interval as part of hospital occupational health care. Serum samples were collected before or on the day of the first vaccine dose (0-day sample, $n = 180$), 16–28 days (mean 20) after the first vaccine dose (3-week sample, $n = 176$), and 13–33 (mean 23) days after the second vaccine dose (34–54 days after the first vaccine dose) (6-week sample, $n = 180$). At enrollment, written informed consent was collected from all participants.

Convalescent phase serum samples ($n = 50$) were collected at HUH from patients with initial RT-qPCR confirmed home-treated COVID-19 infection (Helsinki-Uusimaa health district ethical permission HUS/1238/2020). The patients provided written informed consent and were sampled 14 days–6 weeks after the positive PCR test result. As negative control serum samples ($n = 20$) we used randomly selected diagnostic serum samples collected at TUH prior to COVID-19 pandemic[43]. Negative control serum samples were used for calculation of cut-off values for EIA (see below) and the samples were fully de-identified and collected primarily for epidemiological purposes and thus do not require written informed consent from the subjects.

**Expression and purification of SARS-CoV-2 nucleoprotein and S1 antigens**. SARS-CoV-2 protein expression was done as described previously[43]. Briefly, SARS-CoV-2 N and S sequences were obtained from GenBank (NC_045512.2 and MN908947.3, respectively). N protein was expressed as a fusion protein with glutathione S-transferase (N-GST) in *Spodoptera frugiperda* (Sf-9) cells. GST alone was produced to be used as a control protein. S1 domain of the spike protein (amino acid residues 16–541) was expressed as a fusion protein with mouse IgG2a Fc and 8xhistidine tag (S1-mFc-8xhis) in human embryonic kidney (HEK293F) cells. Mouse promyostatin (ProMstn)-mFc(IgG2a)-6xhis (later referred as Mstn-mFc) was produced to be used as a control protein. Proteins were purified and buffer was exchanged to PBS. Concentrations of produced proteins were measured with BCA protein assay kit (Thermo Fisher Scientific).

**IgG, IgA, and IgM EIA and total Ig EIA for SARS-CoV-2 S1 and N protein antibodies**. Enzyme immunoassay (EIA) was performed as previously described[43] by coating 96-well microtitre plates (Nunc Maxisorp, Thermo Fisher Scientific) for 16 h at $+4\,°C$ with GST-N (2.0 µg/ml), S1-mFc (3.5 µg/ml), and corresponding molar amounts of GST (0.7 µg/ml) and Mstn-mFc (2.4 µg/ml) antigens in PBS. The plates were washed with 0.1% Tween-20 in PBS and blocked for 30 min with assay buffer (5% swine serum (BioInd), 0.1% Tween-20 in PBS) before the addition of 50 µl serum dilutions (final dilution 1:300 or 1:1000 for some analyses in assay buffer). After 2 h incubation at $+37\,°C$, the plates were washed three times followed by the addition of 100 µl horseradish peroxidase (HRP)-conjugated anti-human antibodies (1:8000 dilution of anti-hIgG HRP (Dako A/S), 1:8000 dilution of anti-hIgA HRP (Invitrogen), 1:4000 dilution of anti-hIgM HRP (Dako A/S), and 1:20,000 dilution of anti-hIg (IgG, IgA, IgM) HRP (Abcam)) for 1 h at $+37\,°C$. The plates were washed three times and 100 µl TMB One substrate (Kementec Solutions A/S) was added. The plates were incubated for 20 min at room temperature, 100 µl of 0.2 N sulfuric acid was added to stop the reaction and the levels of IgG, IgA, IgM, and total Ig antibodies were measured at 450 nm with Victor Nivo plate reader (Perkin Elmer). Optical density (OD) values were converted into EIA units by comparing the sample OD values to the OD values of positive (marked as 100) and negative control samples (marked as 0). EIA units <1 were marked as 1. For the present study cut-off units for S1-based EIA were calculated as the average of 20 negative samples plus three standard deviations (SDs) and for N-based EIA as the average of 20 negative samples plus six SDs.

**Propagation of SARS-CoV-2 isolates**. SARS-CoV-2 isolate Fin/25/20 (Gisaid: EPI_ISL_412971) from lineage B.1 was isolated from the nasopharyngeal sample of COVID-19 patient in Finland in February 2020. Swab sample in transport medium was inoculated onto African green monkey kidney epithelial VeroE6 cells at $+37\,°C$ and 5% CO$_2$ in culture medium (Eagle's minimum essential medium (EMEM) supplemented with 2% fetal bovine serum (FBS), 0.6 µg/mL penicillin, 60 µg/mL streptomycin, 2 mM L-glutamine, 20 mM HEPES). Virus was propagated in VeroE6 cells for a total of three times. Subsequently, a VeroE6 clone expressing TMPRSS2, a serine protease essential for SARS-CoV-2 spike protein integrity, was generated (VeroE6-TMPRSS2-H10)[44]. Fin/25/20 was further propagated twice in VeroE6-TMPRSS2-H10 cell line (virus isolate named FIN-25). Another 2020 isolate, SR121 from lineage B.1.463, isolated from a patient in Finland in September 2020, was isolated and propagated only in VeroE6-TMPRSS2-H10 cells[44]. Variants 85HEL of B.1.1.7 lineage and HEL12-102 of B.1.351 lineage were isolated from patients in Finland as described[23] and further propagated only in VeroE6-TMPRSS2-H10 cells. VeroE6-TMPRSS2-H10 cells were maintained in D-MEM (Lonza) supplemented with 10% FBS, 2 mM L-glutamine (Gibco), and penicillin/

streptomycin. For virus propagation in VeroE6-TMPRSS2-H10 cells, D-MEM supplemented with 2% FBS, 2 mM L-glutamine, and penicillin/streptomycin was used. Supernatants containing viruses were harvested, cell debris removed with centrifugation at $500 \times g$ for 5 min, and aliquots stored at $-80\,°C$.

Fifty-percent tissue culture infective dose (TCID$_{50}$) of virus stocks was determined with endpoint dilution assay in VeroE6-TMPRSS2-H10 cells. Briefly, 50,000 cells per well were plated on 96-well tissue culture plates (Sarstedt), and the next day media was changed to infection media (2% FBS). Ten-fold virus dilutions in infection media were applied onto cells, and the plates were incubated for 3 days at $+37\,°C$ and 5% CO$_2$. Cells were fixed with 4% formaldehyde and stained with crystal violet. Virus dilution resulting in 50% cell death was determined to represent TCID$_{50}$ value of the stock virus. Virus propagations and end-point dilution assays were done in BSL-3 laboratory conditions.

**Sequencing of SARS-CoV-2 isolates**. For sequencing of virus stocks, the viral RNA was extracted from supernatants using the RNeasy Mini kit (Qiagen) and reverse-transcribed to cDNA with LunaScript RT SuperMix kit (New England Biolabs). Primer pools (Supplementary Table 1) targeting SARS-CoV-2 were designed using PrimalScheme tool[45] and PCR was done with PhusionFlash PCR master mix (Thermo Scientific). Sequencing libraries were prepared with NEBNext ultra II FS DNA library kit (New England Biolabs) according to the manufacturer's instructions and sequenced using Illumina Miseq with v3 sequencing kit. Raw sequence reads were trimmed, and low quality (quality score <30) and short (<25 nt) sequences were removed using Trimmomatic version 0.36[46]. The trimmed sequence reads were assembled to the reference sequence (NC_045512.2) using BWA-MEM[47] algorithm implemented in SAMTools version 1.8[48]. Sequences of four SARS-CoV-2 isolates used in this study were deposited in GenBank: FIN-25 (GenBank MW717675), SR121 (GenBank MW717676), 85HEL (GenBank MW717677), and HEL-12-102 (GenBank MW717678).

**Illustration of amino acid changes in SARS-CoV-2 spike protein**. SARS-CoV-2 spike structure in closed conformation obtained through cryo-electron microscopy, pdb accession 6VXX[49], was used for illustration of the residue differences between the employed SARS-CoV-2 variants in YASARA (available at http://www.yasara.org/).

**Microneutralization test**. Neutralizing antibodies were measured using a micro-neutralization test (MNT). Serum samples were serially diluted two-fold, starting at 1:10 dilution in 2% FBS in DMEM and incubated with an equal volume of 50 TCID$_{50}$ of SARS-CoV-2 isolate in 96-well tissue culture plates (Sarstedt) for 1 h at $+37\,°C$ (final serum dilution 1:20). VeroE6-TMPRSS2-H10 cells were added (40,000–50,000 cells per well) and the plates were incubated at $+37\,°C$, 5% CO$_2$ for 3 days. Cells were fixed with 4% formaldehyde and stained with crystal violet. MNT titres were calculated as the reciprocal dilution resulting in 50% inhibition of cell death. MNT assays were done at the BSL-3 laboratory conditions.

**Statistical analysis**. Data were analyzed in Excel 2016 (Microsoft). Geometric means with geometric standard deviations (SD) were calculated with GraphPad Prism 8 software. Statistical significance of differences between variants was analyzed with Wilcoxon matched-pairs signed-rank test, and two-tailed $p$-values < 0.05 were considered significant. Differences between age and gender groups were tested with two-tailed Mann–Whitney $U$ test. All serum samples were analyzed in duplicates.

**Reporting summary**. Further information on research design is available in the Nature Research Reporting Summary linked to this article.

## Data availability

All data are available upon request from the corresponding authors. Source data are provided with this paper. The SARS-CoV-2 sequence data generated in this study have been deposited in the Genbank database under accession codes MW717675, MW717676, MW717677 and MW717678.

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

## Acknowledgements
We thank Soili Jussila for cell maintenance, Mikael Ritvos for assistance in protein production, and Anne Suominen, Anne-Mari Pieniniemi, and Simo Miettinen for technical assistance. I.J. was funded by Jane and Aatos Erkko Foundation (grant numbers 3067-84b53 and 5360-cc2fc), the Academy of Finland (AoF; grant numbers 336410 and 337530) and Sigrid Jusélius Foundation. J.H. was funded by AoF (grant number 308613) A.K. was funded by AoF (grant numbers 336439 and 335527), the Finnish Medical Foundation, and private donors through UH. O.V., T.S., and S.K. were funded by Jane and Aatos Erkko Foundation, AoF (grant numbers 336490 to O.V.) and Helsinki University Hospital Funds (TYH2018322). P.A.T., L.I., and J.L. were funded by The Turku University Hospital Research Foundation.

## Author contributions
P.J., L.K., J.L., A.K., and I.J. designed the experiments; P.J., P.K., M.H., S.M., R.L. and L.K. did microneutralization tests and analyzed the data; P.J., A.R. and S.T. did EIA tests and analyzed the data; H.K.H., S.H.P., P.A.T., I.L., A.N., T.M., H.V., L.I., J.L. and A.K. recruited vaccinees and patients and collected their sera and data; A.P., R.N., P.J. and O.R. produced antigens for EIA; P.Ö., S.K., J.H. and O.V. isolated and characterized virus strains; J.H. produced VeroE6-TMPRSS2-H10 cell line; T.S. did sequencing and T.S., J.H. and P.K. analyzed sequences and structures; P.J. analyzed all data sets; P.J., L.K., A.K. and I.J. wrote the manuscript and all co-authors contributed to the edition of the text.

## Competing interests
The authors declare no competing interests.
