## [Peer Review File · Nature Communications]

REVIEWER COMMENTS

Reviewer #1 (Remarks to the Author):

COVID-19 mRNA vaccine induced antibody responses and neutralising antibodies against three SARS

Firstly apologies for not meeting the original deadline.

This is a well conducted longitudinal cohort study in which the authors using in house assays for viral culture, sequencing, serological and micro-neutralising assays evaluate humoral immune responses following the BNT162b2 COVID-19 mRNA vaccination against three SARS-CoV-2 variants. The authors find the vaccination induces effective neutralisation responses against B.1 and SA121 lineages circulating in Finland in spring and autumn 2021, and the B1.1.7 variant whereas neutralising activity is reduced 5 fold for the B.1.351 lineage. Over 90% of vaccinees without a history of prior COVID-19 are expected to have partial protection against the B.1.351 variant. Research team conclude that second dose of BNT162b2 provides efficient cross neutralisation of SARS-CoV-2 variants against current circulating viral variants in Finland.

The result of this paper confirms findings from a number of previous studies. A single dose of the BNT162b2 vaccine induces high neutralising antibody titres in those individual with a previous history of previous COVID-19 infection. Neutralising antibody titres against all four viral isolates exceed that seen in convalescent patient sera recovering from mild disease.

The strengths of the paper are the greater number of study volunteer studies compared to previous studies in a real world setting and the confirmation of previous studies as outlined previously. The numbers of study volunteers allowed the research team to show that the concentration of spike protein antibody (S1 component) decline with age after 2 vaccine shots whereas neutralisation titres were influenced by outcome. Females had slightly higher neutralising activities, although antibody levels were similar to males. Finding related to age, gender and Ig isotype post vaccination have not widely reported to date.

Major Comments

1. Authors should provide evidence as to why a titre 1:20 of is protective in the micro-neutralisation assay. Is this the protection threshold for all four isolates and for all samples or 80% or 50%.

2. Precision of serial dilution (intra and inter assay CV should be mentioned) given that fourfold change in titre us usually associated with a clinically significant change. 95% confidence interval for dilution of 1:20 would be useful given potential overlap between a seronegative and positive vaccine response and the fact that GMT for B 1.351 variant after second dose was B1.351 was only 48.

3. Statement as to overall similarity of cultured B1.351 to original variant should be included as we do not fully know impact of variants within NTD S1 protein or other component of spike protein will have antibody mediated neutralisation

4. The Spearman correlation reflects strength of association between ELISA and neutralisation assay. It is not a measure of agreement between the methods. I am not sure how one would do this here since ELISA values OD values are continuous whereas titration are categorical variables. There may be methods to convert categorical variable to continuous to allow one to utilise Bland Altman and kappa Cohen statistics. Can the authors clarify or get appropriate statistical advice

Minor comments

1. We know that pre-existing immunity to SARS-CoV-2 is associated with protection against re-infection in humans for up to 8 months. The final statement in discussion may in my view be overoptimistic as it overlooks what might be duration of vaccine efficacy against B.1.351 and the possible need for vaccine boosters/modification

2. A comment in the discussion about how representative are health care workers of the total population would be helpful

3. Finding presented in paper unlikely to be applicable to immune compromised patient groups.

4. The authors could make it more clear in the discussion that neutralisation likely to be just one component of protective immunity to SARS-CoV-2 following infection or vaccination

Reviewer #2 (Remarks to the Author):

Jalkanen et al have determined the antibody responses and neutralizing titers against the variants of concern B.1.1.7 (identified first in UK) and B.1.351 (first identified in SA) in 180 health care workers vaccinated with the Pfizer COVID-19 mRNA vaccine 3 weeks after the first dose and 3 weeks after the 2nd dose. They found that after the 2nd dose all individuals had neutralizing antibodies for the B.1.1.7 variant with similar titers to the variant with the D614G substitution, whereas the neutralization of the B.1.351 variant was five-fold reduced. However, most of the vaccinees (92%) had neutralizing antibodies and the geometric mean was higher than that of recovered COVID-19 patients, concluding that the second dose of the vaccine induces efficient cross-neutralization of SARS-CoV-2 variants.

Overall it is a well-written and straightforward manuscript, however, similar data has already been published (some papers already referenced in the manuscript) and therefore it lacks novelty. The study would be of more interest if besides confirmation that levels and neutralization of antibodies elicited by the vaccine are similar for the B.1.1.7 variant and lower for the B.1.351, it would provide additional information like factors behind the variability of antibody responses, studying in more depth the antibody responses in those individuals without neutralizing capacity against the B.1.351, etc.

A strength of the manuscript is the use of live SARS-CoV-2 viruses (isolates) for the neutralization assay.

Some other comments:

- Why the seropositivity cutoff is calculated differently for the S and the N antigens? How was it decided? To our experience 20 pre-pandemic samples may not be enough to establish a reliable cutoff, particularly for N antigens that have high variability and crossreactivity with other coronaviruses.

- What is the performance of the EIA (sensitivity and specificity)? This is relevant for the analysis of seropositive individuals at day 0 or for statements based on N seropositivity for defining SARS-CoV-2 infections.

- Before vaccination (0 day) 11/180 (6%) had anti-S1 IgG antibodies (Supplementary Fig. 1A), however, in Table 1 the frequency of positive responses for anti-S1 IgG is 0% (and 4% for total IgG). Also, the information on IgG N positive responses in d0 in the text and the table is not consistent.

- It would be interesting to highlight in fig 1 the individuals that were seropositive at d0 or at least a comparison of seropositive and seronegative individuals instead of having a separate figure (Supl Fig 1) for the seropositive individuals. As it is shown in the manuscript it is difficult to assess how different are the responses.

- It is curious that no responses of IgA to N in recovered COVID-19 patients. In other and our own studies, IgA N responses are detected even after many months post-acute infection.

- The high variability in IgG S1 responses after the first dose is very interesting. Actually, some individuals have very low IgG levels similar to some recovered COVID-19 patients. However, the 2nd dose seems to compensate and level all individuals. I would add this observation after the following sentence in the Results section. "However, already after the first 97 vaccination dose, the geometric mean of anti-S1 IgG and anti-S1 total Ig antibodies of vaccinees 98 exceeded those of convalescent phase COVID-19 patients"

- Similar to my comment above, it would be better to show the antibody titers against the variants from the previously infected individuals together with non-infected to be able to compare the responses.

- Discussion line 206: It is important to add that neutralizing antibodies were not detectable in all vaccinees.

Minor comments:

-TYKS and HUH abbreviations need to be defined the first time they appear in the manuscript.

Reviewer #1 (Remarks to the Author):

COVID-19 mRNA vaccine induced antibody responses and neutralising antibodies against three SARS

Firstly apologies for not meeting the original deadline.

This is a well conducted longitudinal cohort study in which the authors using in house assays for viral culture, sequencing, serological and micro-neutralising assays evaluate humoral immune responses following the BNT162b2 COVID-19 mRNA vaccination against three SARS-CoV-2 variants. The authors find the vaccination induces effective neutralisation responses against B.1 and SA121 lineages circulating in Finland in spring and autumn 2021, and the B1.1.7 variant whereas neutralising activity is reduced 5 fold for the B.1.351 lineage. Over 90% of vaccinees without a history of prior COVID-19 are expected to have partial protection against the B.1.351 variant. Research team conclude that second dose of BNT162b2 provides efficient cross neutralisation of SARS-CoV-2 variants against current circulating viral variants in Finland.

The result of this paper confirms findings from a number of previous studies. A single dose of the BNT162b2 vaccine induces high neutralising antibody titres in those individual with a previous history of previous COVID-19 infection. Neutralising antibody titres against all four viral isolates exceed that seen in convalescent patient sera recovering from mild disease.

The strengths of the paper are the greater number of study volunteer studies compared to previous studies in a real world setting and the confirmation of previous studies as outlined previously. The numbers of study volunteers allowed the research team to show that the concentration of spike protein antibody (S1 component) decline with age after 2 vaccine shots whereas neutralisation titres were influenced by outcome. Females had slightly higher neutralising activities, although antibody levels were similar to males. Finding related to age, gender and Ig isotype post vaccination have not widely reported to date.

Response: We thank the Reviewer 1 for the scientific view and encouraging comments to further improve our manuscript.

Major Comments

1. Authors should provide evidence as to why a titre 1:20 of is protective in the micro-neutralisation assay. Is this the protection threshold for all four isolates and for all samples or 80% or 50%.

Response: In microneutralization assay, the initial serum dilution was 1:10 and with an equal volume of the virus the final dilution is 1:20. This has now been clarified in the text. 1:20 final dilution was chosen to be the first sample dilution to be analysed since microneutralization

assays are tedious and additional lower antibody dilutions would have consumed a lot of the serum sample. We intend to follow the immune response of the vaccines for several years and thus we have tried to spare the samples for future follow-up studies with potentially novel variants of interest and concern, which are likely to occur in the future.

As stated on lines 383-384 in the Methods section, neutralization titres are calculated as the reciprocal dilution resulting in 50% inhibition of cell death. With 1:20 dilution, we see that 63% of vaccinees have neutralizing antibodies already after the first vaccine dose, indicating that this dilution of serum is working for this method and represents a good compromise. In addition, dilution 1:20 is clearly distinguishable from negative samples collected before vaccination (see figure below). We would like to state that our 1:20 dilution (1:10 serum dilution + an equal volume of virus dilution) equals to 1:10 dilution in the WHO instructions for influenza virus neutralization assays and thus is commonly used dilution factor throughout the World.

2. Precision of serial dilution (intra and inter assay CV should be mentioned) given that fourfold change in titre is usually associated with a clinically significant change. 95% confidence interval for dilution of 1:20 would be useful given potential overlap between a seronegative and positive vaccine response and the fact that GMT for B.1.351 variant after second dose was only 48.

Response: Each neutralization well plate included cell control wells without virus (in duplicates) and virus control wells without serum (in duplicates). Dilution series of serum sample from one COVID-19 patient was included in 10 neutralization plates as positive control. The inter assay CVs were 0% for FIN-25 plates (MNT titre 320), 32% for SR-121 (MNT titre range 320-640) and SA plates (MNT titre range 40-80), and 43% for UK plates (MNT titre range 160-640).

We have now calculated and included the 95% confidence intervals to the Table 1.

3. Statement as to overall similarity of cultured B.1.351 to original variant should be included as

we do not fully know impact of variants within NTD S1 protein or other component of spike protein will have antibody mediated neutralisation

Response: For review purposes, we performed a clade analysis for the variants used in this study and our variants compared very well within the clade with variants isolated in other countries (see figure below). The three virus variants used in the present study represent the viruses of a respective evolutionary lineage/clade and they have been marked as red arrows in the figure. The viruses were also fully sequenced and the sequence information was deposited in the gene bank and thus full data is publicly available for genetic comparisons. Our strains showed the very typical amino acid changes in the S gene that define each of the clades (manuscript Figure 2).

4. The Spearman correlation reflects strength of association between ELISA and neutralisation assay. It is not a measure of agreement between the methods. I am not sure how one would do this here since ELISA values OD values are continuous whereas titration are categorical variables. There may be methods to convert categorical variable to continuous to allow one to utilise Bland Altman and kappa Cohen statistics. Can the authors clarify or get appropriate statistical advice

Response: We agree with the Reviewer that the correlation is not statistically speaking a measure of agreement between two methods but it rather reflects the strength of association. We feel that conversion of the categorical variables (MNT values) into continuous values cannot be done to our dataset. We have now revised the lines 193-197 to emphasize the strength of association instead of agreement. We would like to point out that the correlation coefficient between EIA and MNT were very high, >0.9 in these calculations.

Minor comments

1. We know that pre-existing immunity to SARS-CoV-2 is associated with protection against re-infection in humans for up to 8 months. The final statement in discussion may in my view be overoptimistic as it overlooks what might be duration of vaccine efficacy against B.1.351 and the possible need for vaccine boosters/modification

Response: We have removed the last sentence from the discussion.

2. A comment in the discussion about how representative are health care workers of the total population would be helpful

Response: Healthcare workers represent the Finnish population very well. HCWs in Finland are middle income (nurses) to higher income (doctors) employees. The study subjects were included on a voluntary basis and in the order they received vaccination, no pre-selection was performed. Distribution of age (20-65 years) reflects the vaccinated healthy persons in Finland.

3. Finding presented in paper unlikely to be applicable to immune compromised patient groups.

Response: We fully agree with the Reviewer, but studies of immune compromised patients is out of the scope of this paper. This is a very relevant point and likely many studies in various immune compromised groups are being done in many countries. Those groups are very heterogeneous and thus the reasons for immune deficiencies are different.

4. The authors could make it more clear in the discussion that neutralisation likely to be just one component of protective immunity to SARS-CoV-2 following infection or vaccination

Response: We have already discussed this briefly on lines 273-276 and have referred to two studies on cell-mediated immunity after COVID-19 vaccination. Since we did not study cell-mediated immunity in this report for the sake of clarity we have not extensively discussed the role of cell-mediated immunity. This has to be done in a separate report which concentrates specifically on this topic.

Reviewer #2 (Remarks to the Author):

Jalkanen et al have determined the antibody responses and neutralizing titers against the variants of concern B.1.1.7 (identified first in UK) and B.1.351 (first identified in SA) in 180 health care workers vaccinated with the Pfizer COVID-19 mRNA vaccine 3 weeks after the first dose and 3 weeks after the 2nd dose. They found that after the 2nd dose all individuals had neutralizing antibodies for the B.1.1.7 variant with similar titers to the variant with the D614G substitution, whereas the neutralization of the B.1.351 variant was five-fold reduced. However, most of the vaccinees (92%) had neutralizing antibodies and the geometric mean was higher than that of recovered COVID-19 patients, concluding that the second dose of the vaccine

induces efficient cross-neutralization of SARS-CoV-2 variants.

Overall it is a well-written and straightforward manuscript, however, similar data has already been published (some papers already referenced in the manuscript) and therefore it lacks novelty. The study would be of more interest if besides confirmation that levels and neutralization of antibodies elicited by the vaccine are similar for the B.1.1.7 variant and lower for the B.1.351, it would provide additional information like factors behind the variability of antibody responses, studying in more depth the antibody responses in those individuals without neutralizing capacity against the B.1.351, etc.

A strength of the manuscript is the use live SARS-CoV-2 viruses (isolates) for the neutralization assay.

Response: We thank the Reviewer 2 for the scientific view and comments to further improve our manuscript. As the Reviewer states, the strength of our manuscript is the use of four different live SARS-CoV-2 virus isolates instead of SARS-CoV-2 spike pseudovirus neutralization assay. In addition, the strength of the manuscript is the large number of study subjects from different age and gender groups which enables statistical analyses.

Reviewer 2 suggests that the study would be more interesting if vaccinees with low antibody responses against B.1.351 isolate would be studied more closely. In the present study, we show a relatively high correlation between the neutralization titres against B.1 and B.1.351 virus isolates, and we show that vaccinees with low neutralization titre against B.1 have also low neutralization titres against B.1.351 (Figure 4). According to our experimental data, the main factor behind the low neutralization titres against B.1.351 is the overall low neutralization capacity of individual vaccinee sera against the B.1.351 virus variants. This phenomenon is universal among the vaccinees, which is reflected by the high correlation coefficient in MNT titres between B.1 and B.1.351 variants. Further studies on the individual differences on (any) vaccine responses would require another type of data collection and analyses (e.g. epitope-specific antibody responses), and is not within the scope of this study. This is an important suggestion for future more detailed epitope-specific analyses.

Some other comments:

- Why the seropositivity cutoff is calculated differently for the S and the N antigens? How was it decided? To our experience 20 pre-pandemic samples may not be enough to establish a reliable cutoff, particularly for N antigens that have high variability and crossreactivity with other coronaviruses.

Response: The EIA assay used in this study is described in more detail in our paper "A combination of N and S antigens with IgA and IgG measurement strengthens the accuracy of SARS-CoV-2 serodiagnostics" accepted for publication in the Journal of Infectious Diseases while this manuscript was under review (accepted JID manuscript is now provided with this response). The cut-off values were established again for this paper due to the addition of total

S1-specific Ig measurement, which was not included in the JID paper. Nonetheless, the cut-off values in both papers are comparable. N-protein based EIA had lower variation within control samples and thus a slightly more stringent criteria for cut-off value calculation was chosen.

- What is the performance of the EIA (sensitivity and specificity)? This is relevant for the analysis of seropositive individuals at day 0 or for statements based on N seropositivity for defining SARS-CoV-2 infections.

Response: Differentiation of previously encountered SARS-CoV-2 infection was based on the anti-S1 IgG measurement and not on anti-N measurement. We have changed the sentence on line 83 to "To verify the EIA results and to study the rate of SARS-CoV-2 infections after vaccination, sera were also analyzed with N protein-specific EIA". As stated in the manuscript, the anti-N IgG antibody levels remained at the same level as before vaccination for vaccinees without prior SARS-CoV-2 infection and only one vaccinee developed anti-N IgG antibodies after 0 day sampling. The sensitivity and specificity were not analysed in this study, however, those are analysed in our study describing the EIA method (accepted JID manuscript mentioned above).

- Before vaccination (0 day) 11/180 (6%) had anti-S1 IgG antibodies (Supplementary Fig. 1A), however, in Table 1 the frequency of positive responses for anti-S1 igG is 0% (and 4% for total IgG). Also, the information on IgG N positive responses in d0 in the text and the table is not consistent.

Response: Table 1 describes the results from vaccinees without prior SARS-CoV-2 infection (seronegatives) with anti-S1 IgG antibody measurement. We have clarified this by adding the sentence "without previous SARS-CoV-2 infection" on line 568-569 in Table 1 legend. We are sorry for this unclarity.

- It would be interesting to highlight in fig 1 the individuals that were seropositive at d0 or at least a comparison of seropositive and seronegative individuals instead of having a separate figure (Supl Fig 1) for the seropositive individuals. As it is shown in the manuscript it is difficult to assess how different are the responses.

Response: We feel that Figure 1 would be too complicated with both groups in the same figure. In addition, we feel that the number of samples collected from vaccinees with prior anti-S1 IgG antibodies (n=11) is not large enough to perform reliable statistical analysis and thus the data of seropositive group is separately presented as a supplementary figure.

- It is curious that no responses of IgA to N in recovered COVID-19 patients. In other and our own studies, IgA N responses are detected even after many months post-acute infection.

Response: We agree with the Reviewer and in our studies on hospitalized COVID-19 patients we detect high levels of anti-N IgA antibodies (Jalkanen et al., JID, in press). In this study, we used samples from home-treated COVID-19 patients who suffered from a mild disease. Other

studies have also shown significantly lower IgA levels in mild disease cases compared to a severe disease (<https://www.nature.com/articles/s41591-021-01263-3> and <https://www.ncbi.nlm.nih.gov/pmc/articles/PMC7677074/>). We chose to use home-treated COVID-19 patients in this study since the majority of infections are mild or home treatable and thus they better represent the general COVID-19 patient group. We feel that this type of comparison (vaccines vs. infected) is very important since in the future we have to have a practical vaccination policy also for those individuals who have suffered a previous COVID-19 infection. Even though the vaccination policy is beyond the scope of the present study it may be that one booster dose is sufficient for those individuals who had undergone a confirmed COVID-19 infection.

- The high variability in IgG S1 responses after the first dose is very interesting. Actually, some individuals have very low IgG levels similar to some recovered COVID-19 patients. However, the 2nd dose seems to compensate and level all individuals. I would add this observation after the following sentence in the Results section. "However, already after the first 97 vaccination dose, the geometric mean of anti-S1 IgG and anti-S1 total Ig antibodies of vaccinees 98 exceeded those of convalescent phase COVID-19 patients"

Response: We agree with the Reviewer and have added "despite the initial response after the first vaccine dose" on line 104.

- Similar to my comment above, it would be better to show the antibody titers against the variants from the previously infected individuals together with non-infected to be able to compare the responses.

Response: As stated above, we feel that the figure would be too complicated and the number of previously infected individuals is too small for a reliable statistical comparison.

- Discussion line 206: It is important to add that neutralizing antibodies were not detectable in all vaccinees.

Response: The text has been changed as suggested by the Reviewer, line 210 in this resubmission.

Minor comments:

-TYKS and HUH abbreviations need to be defined the first time they appear in the manuscript.

Response: The abbreviations are now defined and slightly modified to match the hospital English names in the Results section, lines 69 and 70.

REVIEWERS' COMMENTS

Reviewer #1 (Remarks to the Author):

I am very happy with the authors' response and strongly recommend publication.

Reviewer #2 (Remarks to the Author):

Thank you for your responses and for sharing your manuscript describing the EIA (please add the reference in the methods section). Overall the answers are satisfactory and the manuscript has been improved.